# Genome-Wide Identification of Mitochondrial Calcium Uniporter Family Genes in the Tomato Genus and Expression Profilings Under Salt Stress

**DOI:** 10.3390/cimb47121021

**Published:** 2025-12-07

**Authors:** Zhongyu Wang, Jintao Wang, Zepeng Wang, Huifang Liu, Hao Wang, Qiang Wang, Ning Li

**Affiliations:** 1Biological Breeding Laboratory, Xinjiang Uygur Autonomous Region Academy of Agricultural Sciences, Urumqi 830091, Chinawjt_0228@163.com (J.W.); zx1119802932@163.com (Z.W.); huifangliu@xaas.ac.cn (H.L.); 2Vegetable Engineering Technology Research Center, Institute of Fruits and Vegetables, Xinjiang Uygur Autonomous Region Academy of Agricultural Sciences, Urumqi 830002, China; wanghao@xaas.ac.cn

**Keywords:** tomato genus, MCU, evolutionary analysis, salt stress, expression pattern

## Abstract

The mitochondrial calcium uniporter (MCU) is a key channel controlling mitochondrial Ca^2+^ homeostasis, yet its role in plant stress responses remains unclear. Using the tomato pan-genome, this study identified 66 MCU genes across 12 tomato species and grouped them into two distinct evolutionary subfamilies. Phylogenetic, collinearity, and selection pressure analyses revealed that *MCU* genes are evolutionarily conserved and have undergone strong purifying selection. In addition, one MCU gene located on chromosome 6 appears to have originated before the divergence of monocots and dicots, indicating an ancient evolutionary trajectory. Gene structure and conserved motif analyses confirmed their structural conservation, while promoter cis-element analysis suggested that *MCU* genes are widely involved in light and hormone responsiveness. Expression profiling under salt stress showed that multiple *MCU* genes are differentially regulated in a time-dependent manner: *SolycMCU1* and *SolycMCU2* respond rapidly at early stages, whereas *SolycMCU5* and *SolycMCU6* are upregulated during middle and late phases. These results highlight the functional diversification of *MCU* genes in tomato under salt stress. This study provides the first comprehensive evolutionary and functional analysis of the tomato MCU gene family, offering insights into their stress-regulatory mechanisms and potential use in breeding salt-tolerant tomatoes.

## 1. Introduction

Calcium ions function as crucial signaling messengers regulating diverse physiological processes, from nutrient transport to immune and abiotic stress responses [1,2,3]. While the physiological role of MCU has been characterized in model plants like *Arabidopsis thaliana*, focusing on its role in metabolic regulation and stress signaling, its evolutionary history and functional diversity across the Solanum genus remain largely unexplored. Most prior studies relied on a single reference genome, overlooking the structural variations and gene copy number diversity present in wild relatives [4]. In response to fluctuating environmental mineral conditions, plant cells detect changes and initiate specific cytosolic Ca^2+^ oscillations. For example, potassium deficiency triggers root-specific Ca^2+^ signals that activate potassium channels and transporters, thereby enhancing K^+^ acquisition and utilization [5]. The CBL-CIPK signaling module accurately decodes these calcium signals, phosphorylating downstream ion transporters or transcription factors to finely regulate transmembrane ion transport and gene expression, thus maintaining ion homeostasis [6]. Given that salt stress triggers rapid cytosolic Ca^2+^ elevation requiring mitochondrial buffering, we specifically aimed to investigate how MCU evolution contributes to salt tolerance variation.

Calcium signaling serves as a pivotal component within the plant immune system. It plays a crucial role in both pattern-triggered immunity (PTI) and effector-triggered immunity (ETI), as calcium ions are integral to these processes [7]. Upon the detection of pathogen-associated molecular patterns by cell surface receptors, there is a rapid influx of Ca^2+^, resulting in a transient increase in cytosolic Ca^2+^ concentration and the subsequent activation of defense mechanisms [8]. Various pathogens or their effectors can induce unique calcium signatures characterized by specific frequencies and durations, which are interpreted by intracellular calcium sensors to initiate appropriate defense responses [9].

Additionally, in response to abiotic stresses such as drought, high salinity, and extreme temperatures, plants promptly activate calcium signaling pathways [10]. Different abiotic stressors trigger distinct dynamic alterations in cytosolic Ca^2+^. For instance, salt stress activates the SOS pathway in root cells, where the SOS3-SOS2 complex stimulates the plasma membrane Na^+^/H^+^ antiporter SOS1 to expel excess Na^+^ from the cell, thereby maintaining ionic homeostasis [11,12]. Under cold or drought stress, specific calcium signals are perceived by systems such as CBL-CIPK and CDPKs, modulating downstream gene expression and physiological processes such as promoting stomatal closure to reduce water loss or accumulating osmolytes to enhance cellular stress tolerance [13,14,15].

Mitochondria, recognized as the energy center of eukaryotic cells, are pivotal in the conversion of chemical energy obtained from nutrient oxidation into adenosine triphosphate (ATP) through the process of oxidative phosphorylation, thereby facilitating cellular functions [16]. Beyond their role in energy conversion, mitochondria are crucial for the regulation of calcium homeostasis, signal transduction, cellular differentiation, and apoptosis. In response to elevated cytosolic Ca^2+^ levels, mitochondria absorb Ca^2+^ via the mitochondrial calcium uniporter (MCU), a specialized channel situated in the inner mitochondrial membrane. The influx of Ca^2+^ into the mitochondrial matrix activates key enzymes within the tricarboxylic acid (TCA) cycle, subsequently augmenting ATP synthesis. Consequently, the MCU acts as a vital intermediary connecting cytosolic calcium signaling to mitochondrial energy metabolism [17,18]. The MCU, identified in 2011, is responsible for facilitating mitochondrial calcium uptake [19]. Since its discovery, the MCU’s function has been extensively investigated in both animal and human systems. The MCU protein constitutes the central component of a multi-protein complex, with its activity intricately regulated by subunits such as MICU1 and MICU2 to ensure accurate responses to upstream calcium signals. Therefore, the functional state of MCU directly determines the ability of mitochondria to respond to cellular calcium signals, profoundly influencing cellular energy metabolism and signal transduction [20,21].

Research has demonstrated that *MCU* genes are extensively present in plants and share a conserved domain (PF04678) with their animal counterparts [22]. Nonetheless, investigations into plant MCU remain relatively limited. Current studies primarily focus on their roles in calcium homeostasis, signal transduction, growth and development, and responses to abiotic stress. To date, 6, 6, 7, 11, and 4 *MCU* genes have been identified in *Arabidopsis thaliana*, *Zea mays*, pear, *Nicotiana tabacum*, and sorghum, respectively [23,24,25,26]. For instance, the loss of function of *AtMCU1* in *Arabidopsis thaliana* results in abnormal mitochondrial structure, reduced root length, and impaired energy metabolism [27]. Some studies also propose *MCU* involvement in stress responses; for example, under cold stress, the expression of several *GsMCU* genes in wild soybean (*Glycine soja*) changes significantly, suggesting a potential role in cold adaptation [28]. Among the 31 *MsMCU* genes identified in *Medicago sativa,* most exhibited differential expression under salt, drought, and cold stresses, with the exception of *MsMCU1.3*, which did not show a notable response to salt stress [29].

Tomato (*Solanum lycopersicum*) is a significant global crop, yet its production is frequently hindered by abiotic stresses such as high salinity and drought, which adversely affect both yield and quality [30]. Consequently, the identification of stress-related genes in tomatoes is crucial for breeding applications. Wild tomato species serve as valuable genetic resources for enhancing cultivated varieties in particular. For instance, *Solanum pennellii* demonstrates superior adaptive responses to salt stress in comparison to cultivated tomatoes [31]. Additionally, resistance genes identified in *Solanum peruvianum* have contributed essential genetic material for breeding nematode-resistant varieties [32].

However, research on the MCU gene family in tomatoes is limited, especially in terms of systematic identification and evolutionary analysis utilizing pan-genomic resources. To address this research gap, the present study conducts the first genome-wide identification of *MCU* genes across 12 species within the tomato genus. We hypothesized that lineage-specific expansions or losses of *MCU* genes during Solanum evolution have driven the functional diversification of calcium handling, thereby contributing to the differential salt stress tolerance observed between cultivated and wild tomatoes. This study provides an important foundation for further elucidating the regulatory network and biological functions of MCU in tomato, thereby supporting future efforts in stress-resilient crop breeding.

## 2. Materials and Methods

### 2.1. Plant Materials and Stress Treatments

The tomato cultivar Heinz 1706 (*Solanum lycopersicum* cv. Heinz 1706) was used in this study. The seeds were sourced from the Bio-breeding Laboratory, Xinjiang Academy of Agricultural Sciences. Following surface sterilization with 0.1% HgCl_2_ for a duration of 10 min, the seeds underwent five rinses with sterile water and were subsequently germinated on Petri dishes lined with filter paper within a growth chamber maintained at 25 ± 1 °C. Seedlings exhibiting uniform height, leaf area, and growth vigor at the four-true-leaf stage (approximately 25 days old) were selected for experimental procedures. For salt stress treatment, the soil-growing seedlings were irrigated with 200 mM NaCl solution until the soil was saturated, and the flowerpots were placed on trays to retain the excess solution. Samples were collected at five distinct time points: 0 h, 1 h, 2 h, 6 h and 12 h. The samples taken at 0 h were used as the control group. Each treatment condition was replicated biologically three times. Approximately 500 mg of fresh leaf tissue was harvested per replicate, immediately flash-frozen in liquid nitrogen, and stored at −80 °C. A subset of the samples was allocated for RNA extraction, while the remaining portion was reserved for subsequent physiological and biochemical analyses.

### 2.2. Identification of the MCU Gene Family in Tomato

The genome (SL4.0) and protein sequences of the cultivated tomato Heinz 1706 were downloaded from the JGI Data Portal (https://data.jgi.doe.gov/, accessed on 24 July 2025). MCU protein sequences of *Arabidopsis thaliana* were sourced from the TAIR database (https://www.arabidopsis.org/, accessed on 29 April 2025) and employed as query templates. Genomic and protein sequences of the wild tomato were acquired from previous sequencing and assembly efforts conducted by our research group [33]. The BLASTP (v2.16.0) was performed (E-value ≤ 1 × 10^−10^) for comprehensive protein alignment. A domain search was performed using the MCU domain model (PF04678) from the Pfam database (http://pfam-legacy.xfam.org/, accessed on 2 July 2025) via HMMER_3.3.2 software (E-value ≤ 1 × 10^−5^). The final *MCU* gene family members were identified by intersecting the results from BLASTP (v2.16.0) and HMMER (v3.3.2) analyses. Gene annotation information was extracted from GFF3 files.

### 2.3. Analysis of Protein Physicochemical Properties and Structure

The molecular weight (MW), isoelectric point (pI), grand average of hydropathicity (GRAVY), and amino acid composition of MCU proteins were calculated utilizing the Expasy ProtParam tool (available at https://www.expasy.org, accessed on 17 July 2025). The secondary structural elements of proteins, including α-helices, β-sheets, β-turns, and random coils, along with their proportional distributions were predicted using the SOPMA module on the NPS server (available at https://npsa-prabi.ibcp.fr/, accessed on 24 July 2025).

### 2.4. Chromosomal Localization and Phylogenetic Analysis

The chromosomal locations of *MCU* genes were determined based on the cultivated tomato genome annotation file, employing the Chromosome Position Plotting module in TBtools v2.336 [34]. Multiple sequence alignment (MSA) was conducted using MUSCLE with default settings. A maximum likelihood (ML) phylogenetic tree was generated using IQ-TREE_2.4.0 with the best-fit model (WAG + G4) selected automatically and 1000 bootstrap replicates. The phylogenetic tree was visualized and annotated using iTOL (available at https://itol.embl.de/, accessed on 20 August 2025). Gene structure diagrams were produced using the Gene Structure Display Server module in TBtools v2.336.

### 2.5. Gene Collinearity and Ka/Ks Analysis

A cross-species collinearity analysis was executed utilizing the OneStepMCScanX-SuperFast module within TBtools (v2.336) (evalue = 1 × 10^−5^, blasthit = 10), to compare *MCU* genes across various species, including *Sorghum bicolor*, *Oryza sativa*, *Arabidopsis thaliana*, and *Glycine soja*. Intraspecific collinearity analysis was conducted through pairwise comparison to identify homologous gene clusters. The YN model in KaKs_Calculator_2.0 was employed to calculate the non-synonymous (Ka) and synonymous (Ks) substitution rates for homologous gene pairs, as well as their Ka/Ks ratio, to evaluate selective pressures during evolution.

### 2.6. Cis-Acting Element Analysis

Promoter sequences extending 2000 bp upstream of the translation start site of *MCU* genes were extracted using the getfasta tool in BEDTools (v2.31.1). Putative cis-acting regulatory elements were predicted using the PlantCare database (http://bioinformatics.psb.ugent.be/webtools/plantcare/html/, accessed on 4 September 2025). The identified cis-elements were counted and assigned to four functional categories: Growth, Stress, Phytohormone, and Light-according to their annotated functions in PlantCARE. The numbers of each cis-element type in the promoters were visualized as a heatmap using the HeatMap module in TBtools (v2.336) with default parameters. In the heatmap, cis-elements are ordered according to these predefined functional categories, and the color intensity represents the abundance of each cis-element in a given promoter; no hierarchical or other unsupervised clustering algorithm was applied.

### 2.7. Transcriptome Analysis and Expression Profiling

Publicly available RNAseq data for roots of the cultivated tomato M82 and the salt-tolerant wild species *S. pennellii* subjected to NaCl treatment for 12 h were obtained from our previous study [35]. In that study, raw reads were aligned to the tomato reference genome and gene expression levels were quantified as FPKM. In the present work, we directly used these published FPKM values of *MCU* genes. The Heatmap module within the TBtools v2.336 software package was employed to perform log2(FPKM) processing and data standardisation (Z-score transformation), generating an expression heatmap.

### 2.8. Protein–Protein Interaction Network Prediction

The MCU protein sequences from cultivated tomatoes were submitted to the STRING database (https://cn.string-db.org/, accessed on 21 August 2025) for interaction prediction, employing a moderate confidence threshold (≥0.4). The resulting network was analyzed and visualized using Cytoscape v3.9.1.

### 2.9. Quantitative Real-Time PCR Validation

Gene-specific primers were designed based on unique and divergent regions of each *SolycMCU* gene sequences (Table 1), with Actin employed as the internal reference gene. Total RNA was isolated utilizing the TIANGEN DP441 Plant RNA Kit according to the manufacturer’s instructions. Subsequent reverse transcription was conducted using the abm All-In-One 5× RT Master Mix in a 20 µL reaction system, comprising 1 µL of cDNA template and 0.4 µL each of forward and reverse primers. The qPCR amplification was executed using the Vazyme ChamQ Universal SYBR qPCR Master Mix under the following thermal cycling conditions: an initial denaturation at 94 °C for 120 s, followed by 45 cycles of 94 °C for 5 s, 58 °C for 15 s, and 72 °C for 10 s. Three biological replicates were included per sample. Relative expression levels were determined employing the 2^^(−ΔΔCT)^ method. The primers were designed based on conserved regions of *MCU* genes and validated for specificity via Primer-BLAST (https://www.ncbi.nlm.nih.gov/tools/primer-blast/, accessed on 10 September 2025). Statistical analysis was performed using one-way ANOVA in GraphPad Prism 8.0.2, with a threshold for statistical significance set at *p* < 0.05.

## 3. Results

### 3.1. Analysis of MCU Gene Family Members in Tomato

A comprehensive genome-wide analysis encompassing 12 tomato species and the outgroup *S. lycopersicoides* identified a total of 66 members of the *MCU* gene family (Table 2 and Appendix A). Within this group, the outgroup *S. lycopersicoides* and the wild species *S. peruvianum* each contained 4 *MCU* genes, *S. pennellii* contained 5, and the remaining species each possessed 6 *MCU* genes. In the cultivated tomato Heinz 1706, the protein SolycMCU2 was identified as the smallest, comprising 176 amino acids (aa) and having a molecular weight of 20,541.07 Da, whereas SolycMCU6 was the largest (365 aa, 41,922.88 Da). The isoelectric points (pI) of the MCU proteins spanned from 6.97 (SpenMCU2) to 9.66 (ScorMCU4). Predictions of subcellular localization suggested that among the MCU proteins in the Heinz tomato, only SolycMCU2 was localized to both chloroplasts and mitochondria; all others were predicted to be mitochondrial. All MCU proteins were predicted to be hydrophilic, as indicated by negative GRAVY values. Secondary structure analysis indicated that α-helices and random coils were the predominant structural motifs. Comparative analysis within the tomato clade revealed that SlydMCU1 had the fewest amino acids (92 aa), whereas SolycMCU6 had the most (365 aa), suggesting potential functional divergence associated with sequence length variation.

### 3.2. Chromosomal Localization of MCU Genes in Cultivated Tomato

The chromosomal localization of the 6 *MCU* genes in the cultivated tomato Heinz 1706 (Figure 1) demonstrates a non-uniform distribution: Chr2 and Chr3 each contained 2 genes, whereas Chr4 and Chr6 each contain one gene. Notably, *SolycMCU2*, *SolycMCU5*, and *SolycMCU6* are situated on the long arms of their respective chromosomal, while *SolycMCU3* and *SolycMCU4* are located on the short arms. Analysis of wild tomato species reveals a largely conserved chromosomal distribution pattern for *MCU* genes (Appendix A). However, it is significant that *Solanum peruvianum* and *Solanum lycopersicoides* lack MCU genes on Chr3, consistent with lineage-specific gene loss events during the evolutionary divergence of the Solanum genus.

### 3.3. Phylogenetic Analysis of MCU Genes in Tomato

A maximum likelihood phylogenetic tree was constructed using 6 *A. thaliana* and 66 tomato MCU protein sequences (Figure 2) to elucidate the evolutionary relationships within the *MCU* family. Based on evolutionary distances and the established classification of *A. thaliana* MCU subfamilies, coupled with domain features of tomato MCUs, all identified genes were classified into two distinct subfamilies: Group I comprised 2 *AtMCU* genes and 12 tomato *MCU* genes, all of which were located on chromosome 6. Group II consisted of 4 *AtMCU* genes and 54 tomato *MCU* genes, which were distributed across multiple chromosomes. Genes located on the same chromosome in both *A. thaliana* and tomato tended to cluster within the same clade with high sequence homology, indicating that these are orthologous genes derived from a common ancestral gene prior to speciation, thereby underscoring the genetic conservation of *MCUs* during evolution. A comparison between cultivated and wild tomatoes (*S. peruvianum*, *S. lycopersicoides*, *S. chilense*, *S. pennellii*) revealed that while cultivated tomato possesses a complete set of 6 *MCU* genes distributed across both subfamilies, wild species commonly lack specific *MCU* genes on Chr3. The wild species exhibiting gene deletions on Chr3 are positioned on more basal branches of the phylogenetic tree, indicating a comparatively ancient evolutionary lineage. This distinct distribution pattern reflects the divergence in evolutionary history between these wild lineages and the cultivated tomato, indicating lineage-specific variation in gene retention or loss.

### 3.4. Gene Structure and Conserved Motif Analysis of Tomato MCU Genes

The analysis of gene structure and conserved motifs has uncovered substantial compositional differences and potential functional divergence among the tomato *MCU* members (Figure 3). Notably, *SolycMCU3* and *SolycMCU4* lacked Untranslated Regions (UTRs), a structural characteristic that may affect their post-transcriptional regulation. Further motif analysis indicated that the distinction between *SolycMCU3* and *SolycMCU4* is solely attributed to the presence of motif7 in *SolycMCU4*, implying that motif7 may impart functional specificity. Conversely, *SolycMCU1* and *SolycMCU5* each possess an additional motif9 compared to *SolycMCU3*/*4*, although their positional difference may influence function by affecting protein structure or interactions. *SolycMCU6* displays a unique motif configuration with additional motif10 and motif8 at its 5′ and 3′ termini, respectively, potentially conferring more intricate regulatory capabilities. Importantly, *SolycMCU2* retains only motifs 8, 2, 1, and 4, a configuration markedly distinct from other members, which may represent the core motifs essential for its fundamental function.

The *MCU* gene family exhibits significant conservation across tomato species, yet it also presents structural variations. With the exception of *SgalMCU3* from a related species, all *MCU* genes are characterized by the presence of two exons. Notably, *MCU* genes located on Chr1 (e.g., *ScorMCU1* and *SneoMCU1* in *Solanum corneliomulleri*, *Solanum neorickii*, respectively) had significantly fewer motifs and frequently lack untranslated regions (UTRs) compared to their counterparts in other clusters. This observation may be indicative of gene simplification or functional reduction. For instance, *SolycMCU2* and *SlydMCU2* contain only motifs 8, 2, 1, and 4, whereas *SpenMCU2* lacked motif6 and motif10. These variations in motif composition could influence synergistic interactions with other *MCU* genes. Despite these variations, motifs 1 and 4 were highly conserved across all *MCU* genes, indicating their essential functional role, likely associated with fundamental processes such as calcium ion homeostasis. Considering the diverse ecological niches occupied by these species, the convergent structural simplification observed on specific chromosomes may represent a significant evolutionary pattern in the adaptation of the tomato clade to varying environmental conditions.

### 3.5. Collinearity and Evolutionary Pressure Analysis of the Tomato MCU Gene Family

Intraspecific collinearity analysis was conducted to generate synteny maps of *MCU* homologous genes within tomato species (Figure 4A). In the cultivated tomato (*S. lycopersicum*), syntenic pairs were identified between *SolycMCU1* and *SolycMCU3*, as well as between *SolycMCU1* and *SolycMCU5*. Conversely, the wild tomato *S. lycopersicoides* displayed only a single syntenic pair (*SlydMCU2*/*SlydMCU3*), highlighting differences in *MCU* gene duplication events among species. Notably, with the exception of *S. lycopersicoides*, other tomato accessions exhibited *MCU* synteny across chromosomes, suggesting that gene family expansion occurred through whole-genome or segmental duplication events.

Interspecific collinearity analysis among *Arabidopsis*, tomato, wild soybean (*Glycine soja*), rice (*Oryza sativa*), and *Sorghum bicolor* identified only one syntenic *MCU* gene pair between monocots and dicots (Figure 4B). This finding indicates substantial evolutionary divergence within the *MCU* family following the separation of monocot and dicot lineages. Notably, the *SolycMCU6* gene on tomato chromosome 6 exhibits features suggesting an ancient origin relative to the monocot-dicot divergence, as supported by phylogenetic clustering and its syntenic relationship with species such as rice and *Arabidopsis*. This gene is inferred to originate from a common ancestor of monocot *MCU* genes and has been retained through a unique evolutionary trajectory in tomato, implying a potential specialized function.

The analysis of chromosomal distribution within the tomato clade demonstrated substantial conservation in the loci of most MCU genes. Notably, the absence of two MCU genes on chromosome 3 in *S. lycopersicoides* and *S. peruvianum* may be attributed to with genomic segment loss events throughout evolutionary history. Phylogenetic analysis indicated that MCU genes located on Chr6 were more conserved compared to homologous genes on other chromosomes, whereas those on Chr2, Chr3, and Chr5 displayed relatively conserved sequences (Figure 4C).

The calculation of the non-synonymous to synonymous substitution rate (Ka/Ks) for intraspecific syntenic homologous gene pairs resulted in Ka/Ks values less than 1 for all pairs analyzed. This finding suggests that the *MCU* gene family in tomatoes has been subject to strong purifying selection over the course of evolution, thereby preserving functional conservation. This is likely due to the essential role of MCU proteins in mitochondrial calcium transport, where any functional impairment could potentially disrupt cellular energy metabolism.

### 3.6. Cis-Acting Element Analysis of Tomato MCU Genes

A systematic investigation of cis-acting regulatory elements (CREs) was performed within the promoter regions (approximately 2000 bp upstream of the translation initiation site) of the 66 *MCU* genes (Figure 5). Using the PlantCARE database, 47 distinct CRE sites were identified. The light-responsive element Box 4 (5′-ATTAAT-3′) emerged as the most prevalent, detected in 66 promoters, which implies a potential central function in light signal transduction. Other significantly enriched elements included the G-box (5′-CACGTG-3′), ABRE (5′-ACGTG-3′; involved in abscisic acid (ABA) signaling), and the CGTCA- and TGACG-motifs (associated with methyl jasmonate (MeJA)-mediated defense responses). Functional categorization demonstrated a pronounced bias in CRE distribution, with light-responsive and hormone-responsive elements collectively constituting approximately 63.8% of the total. This finding suggests that the MCU gene family might play a key role in modulating plant growth and responses to abiotic stress by integrating light, and phytohormone signaling pathways.

### 3.7. Transcriptome Analysis of Cultivated Tomato M82 and Wild Tomato S. pennellii Under Salt Stress

A comparative analysis of *MCU* gene expression patterns between the cultivated tomato M82 and the wild tomato *S. pennellii* under salt stress conditions revealed distinct expression dynamics. Under control conditions (Figure 6), *MCU3* exhibited low expression levels in the root tissues of both genotypes, with no significant induction observed in response to salt stress. The expression profiles of other MCU family members demonstrated marked species-specific differences. In the absence of stress, *MCU1*, *MCU5*, and *MCU6* were expressed at significantly higher levels in M82 compared to *S. pennellii* (*p* < 0.05), whereas *MCU2* expression was comparable between the two genotypes.

Upon exposure to 200 mM NaCl, *S. pennellii* showed a significant upregulation in the expression of *MCU1*, *MCU2*, *MCU4*, and *MCU5* relative to control conditions (*p* < 0.05), with fold changes of 2.23 and 2.11 for *MCU1* and *MCU4*, respectively. In contrast, in the cultivated M82, only *MCU2* exhibited a significant response to salt stress, with a 2.13-fold change. Notably, *MCU6* expression was significantly downregulated in both genotypes under stress conditions, with fold changes of 2.32 in M82 and 1.76 in *S. pennellii* (*p* < 0.05). Based on the expression profiles, the expression intensity of *MCU1*, *MCU2*, and *MCU4* genes exhibited a positive correlation with salt stress treatment, suggesting their potential involvement in the salt stress response. These differential expression patterns may reflect the stronger salt tolerance of wild tomato, potentially mediated through a more robust calcium signaling response via *MCU* genes to maintain cellular homeostasis under stress. However, specific functional mechanisms require systematic validation using gene editing techniques coupled with physiological and biochemical assays.

### 3.8. Prediction of the MCU Protein Interaction Network in Cultivated Tomato

The prediction of protein-protein interaction networks utilizing the STRING database (Figure 7) revealed that *SolycMCU5* does not have significant interactors among the *SolycMCUs*. Conversely, an analysis of interaction profiles from the UniProt database suggests that other SolycMCU proteins may interact with a variety of functional proteins, including those containing EF-hand domains, FYVE-type zinc finger domains, Band-7 family transmembrane domains, and Prohibitin family proteins. The EF-hand domain is recognized as a classical calcium-binding motif, and proteins encoding this domain can be classified into two categories: those involved in calcium signal transduction (e.g., Calmodulin, Calnexin) and those serving as calcium buffers to maintain cellular calcium homeostasis (e.g., Caldesmon, Calretinin). Importantly, EF-hand domain proteins are integral to the regulation of mitochondrial Ca^2+^ uptake and the maintenance of Ca^2+^ homeostasis. By integrating the interaction network data from this study with the structural and functional attributes of EF-hand proteins, we hypothesize that SolycMCU proteins may form complexes with calcium-binding proteins. This cooperation could facilitate the precise regulation of the mitochondrial calcium uniport process, thereby fine-tuning the mitochondrial calcium uptake rate and contributing to the maintenance of cellular calcium homeostasis.

### 3.9. Quantitative Real-Time PCR (qRT-PCR) Analysis of Tomato MCU Genes

To investigate the expression patterns of the mitochondrial calcium uniporter (MCU) family members under salt stress and their potential roles in the stress response, the spatiotemporal expression profiles of *MCU* genes in cultivated tomato were validated using qRT-PCR (Figure 8). Among the six identified *MCU* homologous genes, expressions of *SolycMCU3* and *SolycMCU4* were undetectable in leaf tissue, a finding corroborated by transcriptome data from the Tomato Functional Genomics Database (TFGD; http://ted.bti.cornell.edu/cgi-bin/TFGD/digital/home.cgi, accessed on 12 September 2025). The remaining four genes exhibited distinct temporal expression patterns, suggesting roles at different stages of salt stress.

The relative expression of *SolycMCU1* was significantly upregulated, peaking at 2 h of salt stress, while its expression remained low and unchanged at other time points (0 h, 1 h, 6 h, 12 h). This rapid induction during the early stage (2 h) implies that *SolycMCU1* might participate in the initial tomato response to salt stress, potentially by modulating mitochondrial calcium homeostasis to facilitate rapid stress signaling or initiate metabolic adjustments. *SolycMCU2* expression also increased significantly after 2 h of stress but decreased markedly at 6 h and 12 h, falling below levels observed at 0 h and 1 h. This temporal pattern suggests *SolycMCU2*’s involvement is confined to the early stress response phase (around 2 h), likely contributing to initial signal transduction alongside *SolycMCU1*. *SolycMCU5* exhibits a distinctive expression pattern, with transcription levels peaking at 2 h (1.12-fold increase) and showing significant declines at both 1 and 6 h. At 12 h, expression levels were slightly below pre-treatment levels. This may indicate the gene plays a role in adaptive responses during prolonged stress processes. Notably, *SolycMCU6* showed a dynamic response: expression began to increase at 1 h, reached its peak at 6 h, and then declined significantly by 12 h, falling below the 0 h level. Its mid-phase peak (6 h) suggests *SolycMCU6* might function during sustained stress exposure, potentially enhancing tomato salt tolerance by optimizing mitochondrial calcium homeostasis and participating in processes like energy metabolism and reactive oxygen species (ROS) scavenging. The decline at 12 h could result from accumulated cellular damage under prolonged stress or a shift in regulatory focus to other adaptation mechanisms. The expression profiles of *SolycMCU1*, *SolycMCU2*, *SolycMCU5*, and *SolycMCU6* indicate their potential involvement in the early response, mid-term regulation, and sustained adaptation to salt stress, respectively. This provides transcriptional evidence for further elucidating the functional division within the *MCU* family during the tomato salt stress response.

## 4. Discussion

The mitochondrial calcium uniporter (MCU) is integral to calcium ion uptake in plant mitochondria and is pivotal in cellular calcium signaling. Through analysis of the tomato pan-genome, this study identified 66 *MCU* genes across 12 species within the Solanum genus. Evolutionary analysis classified these genes into two conserved subfamilies, a classification that aligns closely with observations in species such as *A. thaliana*, maize (*Zea mays*), and pear. For instance, recent work on the pear genome similarly identified seven *PbrMCU* genes classified into two subgroups, with segmental duplication identified as a primary driving force for their expansion [24,25]. This suggests an ancient and deeply conserved evolutionary history for *MCU* function within the plant kingdom. Notably, synteny analysis revealed that only *SolycMCU6* in cultivated tomato shared a collinear relationship with *MCU* genes in the monocot rice (*Oryza sativa*), but not with those in *A. thaliana*. This distinct collinearity pattern suggests that *SolycMCU6* may represent an ancient lineage retained in tomato, implying a potential unique functional role or evolutionary history distinct from the *A. thaliana* homologs. Furthermore, Ka/Ks analysis indicated that all *MCU* genes have been subjected to strong purifying selection, underscoring the essential role of their encoded proteins for tomato survival and the high evolutionary constraint on their core sequences and functions.

Gene duplication is a fundamental evolutionary mechanism driving the expansion and functional diversification of gene families. Our investigation revealed that the chromosomal locations of *MCU* genes are highly conserved within the Solanum genus. Moreover, most genes were positioned distally from centromeres, in genomic regions typically associated with more open chromatin structures, which may provide a favorable environment for duplication events. Intraspecific collinearity analysis in tomato further supported gene expansion via duplication. However, both intra-genus and interspecies analyses indicated that the *MCU* gene family overall exhibits a high degree of evolutionary conservation.

Analysis of gene structure and functional motifs unveiled the interplay between functional conservation and potential regulatory diversity among *MCU* genes. Gene structure analysis showed that tomato *MCU* genes contain only 2–3 exons, a configuration that is highly consistent with that observed in species like *Arabidopsis*, pear, and maize, indicating stable genomic organization across diverse plant species. Conserved motif analysis revealed that while minor gains or losses of specific motifs occurred among *MCU* genes from different species, the core set of motifs remained largely conserved. To enhance our understanding of their regulatory mechanisms, we predicted cis-acting regulatory elements within the 2000 bp promoter region upstream of all *MCU* genes. The results demonstrated that these promoters harbor a rich diversity of elements. Light-responsive elements were the most abundant in both type and number, followed by hormone-responsive elements, primarily associated with abscisic acid (ABA) and methyl jasmonate (MeJA). These predictions suggest that tomato *MCU* gene expression might be precisely regulated by light and hormone signals, potentially fine-tuning mitochondrial calcium transport. Significantly, both ABA and MeJA are pivotal hormones in plant responses to abiotic stress, implying that *MCU* genes could integrate mitochondrial calcium signaling into the broader regulatory network governing stress adaptation. This inference is further supported by protein interaction predictions, which indicated that MCUs may interact with several EF-hand domain-containing proteins (e.g., A0A3Q7EXD5, A0A3Q7GPL6), FYVE-type domain proteins (e.g., A0A3Q7GBL4, A0A3Q7EYY4), and Band 7 family proteins (A0A3Q7IB34), all of which are implicated in calcium binding or homeostasis maintenance. This suggests that MCUs may operate collaboratively with other calcium-signaling proteins to form a sophisticated regulatory module for calcium transport.

To validate the predictions from the bioinformatic analyses, we analyzed the expression patterns of *SolycMCU* genes in cultivated tomato under salt stress using qRT-PCR, and further investigated the expression profiles of MCU family members using root expression data from cultivated tomato M82 and wild tomato *S. pennellii*. Although these datasets reflect responses in distinct tissues and genetic backgrounds, which limits the ability to make direct quantitative comparisons between the leaf and root expression profiles, the RNA-seq data provides independent, qualitative confirmation that *MCU* genes are actively regulated under salt stress in tomato roots. Taken together with the leaf qRT-PCR results, these findings suggest that transcriptional regulation of *MCU* genes is a common feature of the salt stress response in tomatoes, although the specific temporal and magnitude dynamics likely vary between tissues and genotypes. *SolycMCU1* and *SolycMCU2*, acting as early responders, are likely involved in the initial perception and transduction of the stress signal; *SolycMCU5*, a mid-phase respondent, may function during the transition to stress adaptation, whereas *SolycMCU6*, as a late responder, might contribute to the establishment and maintenance of long-term salt tolerance. Additionally, we found that *SolycMCU3* and *SolycMCU4* were not expressed in tomato leaves. Notably, these two genes were lost on chromosome 3 in *S. pennellii* and *S. lycopersicoides*, and one of each was lost in *S. chilense* and *S. pennellii*, while being retained in other closely related wild species. We propose that the absence of *SolycMCU3* and *SolycMCU4* in specific species reflects lineage-specific gene loss events during the evolutionary history of the Solanum genus. In species where these genes are retained, they may contribute to specific physiological functions, potentially including tissue-specific expression in organs such as roots or fruits, subject to further functional verification. Future research should prioritize validating the tissue-specific expression patterns of these two genes, which is crucial for a comprehensive understanding of plant MCU protein function.

In conclusion, this study integrates bioinformatics and experimental validation to systematically elucidate the evolutionary characteristics, structural conservation, and potential functional roles of the tomato *MCU* gene family in response to salt stress. It is noteworthy that analysis of root transcriptome data from salt-stressed cultivated tomato M82 and wild tomato *S. pennellii* revealed more complex expression patterns and higher expression levels of *MCU* genes in the wild species. Given the superior salt tolerance of *S. pennellii* compared to cultivated tomato, we hypothesize that differential expression patterns of *MCU* genes across tomato species and tissues could lead to variations in calcium recognition and transport efficiency, ultimately influencing the overall salt tolerance of the plant. A more comprehensive profiling of *MCU* gene expression in wild tomato species under salt stress in future studies will provide stronger support for this hypothesis and offer a critical theoretical foundation for utilizing superior *MCU* alleles from wild genetic resources to improve salt tolerance in cultivated tomato.

## 5. Conclusions

This study demonstrates that the mitochondrial calcium uniporter (MCU) gene family in tomato is evolutionarily highly conserved. Phylogenetic analysis classified its members into two stable subfamilies, with all genes undergoing strong purifying selection, underscoring their essential role in maintaining cellular calcium homeostasis and mitochondrial function. Analyses of gene structure, conserved motifs, and cis-acting regulatory elements further revealed both conservation and diversity in their sequences and regulatory mechanisms. Notably, the promoter regions were enriched with light-responsive and hormone-responsive elements, particularly to ABA and MeJA, suggesting that *MCU* genes may integrate light and stress-related hormonal signals to participate in adaptive responses to environmental stress. Expression profiling confirmed distinct temporal response patterns of *MCU* genes under salt stress. The evolutionary loss or altered expression of *SolycMCU3* and *SolycMCU4* likely reflects lineage-specific evolutionary processes or functional diversification. Future research should focus on the following two aspects: firstly, verifying the tissue-specific expression and function of *SolycMCU3* and *SolycMCU4* to clarify their potential neofunctionalization or redundancy during tomato evolution; secondly, functionally characterizing superior MCU alleles identified in wild species (e.g., through gene editing or overexpression). These efforts will provide essential theoretical insights for leveraging wild genetic resources to improve salt tolerance in cultivated tomatoes. In summary, the MCU gene family plays a critical role in tomato adaptation to salt stress, and the functional divergence among its members provides a theoretical basis and genetic resources for improving salt tolerance in cultivated tomato using wild relatives. Future studies should focus on elucidating the regulatory networks of MCU proteins and their functional mechanisms across different tissues to inform new strategies for breeding stress-resistant crops.

## Figures and Tables

**Figure 1 cimb-47-01021-f001:**
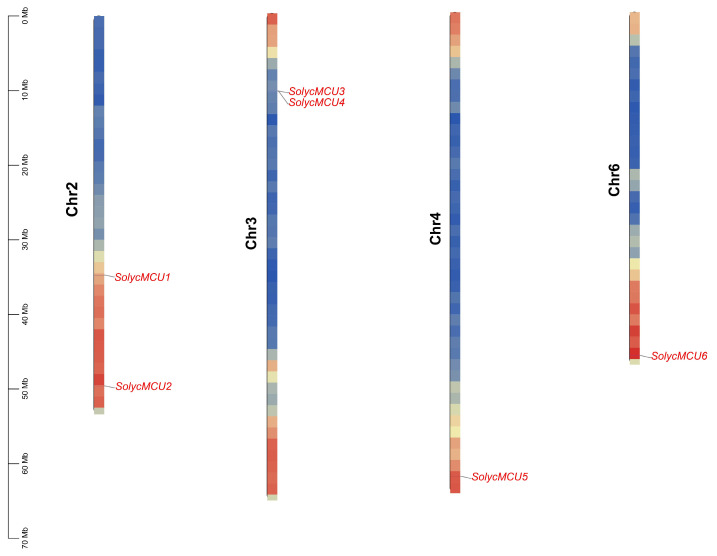
The position of the MCU genes on the chromosomes of cultivated tomato. The color gradient on the chromosome ranges from red to blue, representing gene density from high to low. Red signifies regions of high gene density, while blue represents areas of low gene density.

**Figure 2 cimb-47-01021-f002:**
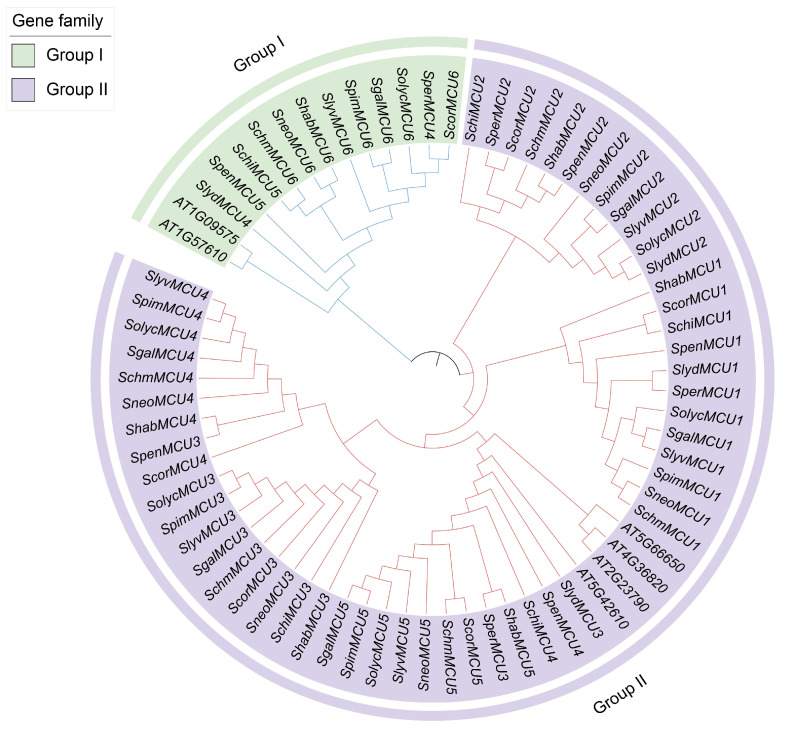
Phylogenetic tree of the MCU gene family in tomato genus. Different color represent different subfamily, green and purple represent the Group I and II, respectively.

**Figure 3 cimb-47-01021-f003:**
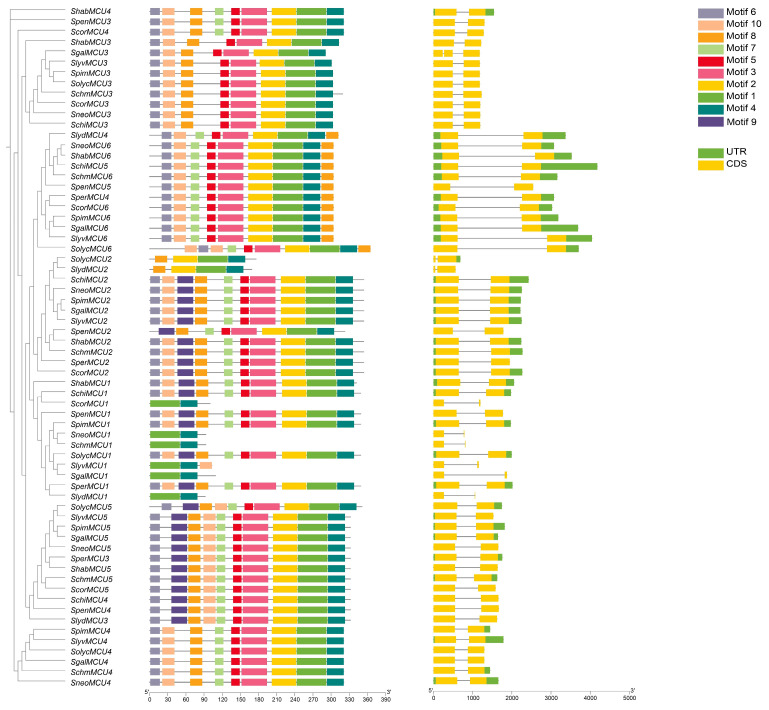
Genetic structure and conserved motifs of MCU genes in tomato genus.

**Figure 4 cimb-47-01021-f004:**
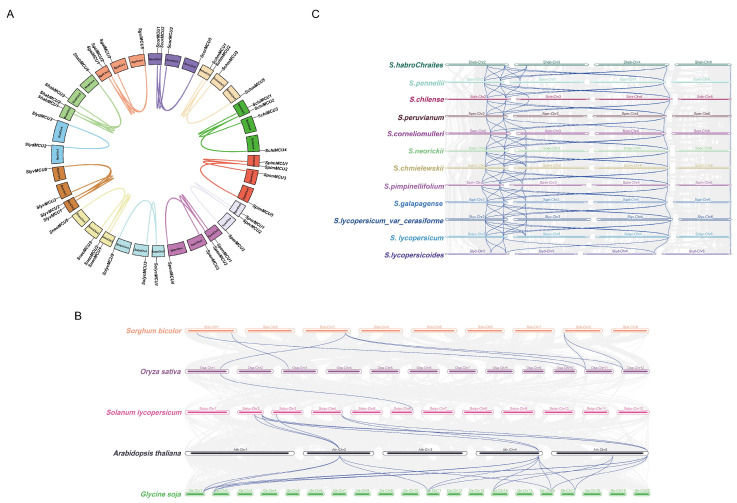
MCU genes colinearity analysis. (**A**). Homologous gene collinearity analysis in tomato species. (**B**). Collinearity analysis of the *MCU* genes in *Sorghum biocolor*, *Oryza sativa*, *Solanum lycopersicum*, *Arabidopsis thaliana*, and *Glycine soja*. (**C**). Collinearity analysis of *MCU* Genes within the tomato genus.

**Figure 5 cimb-47-01021-f005:**
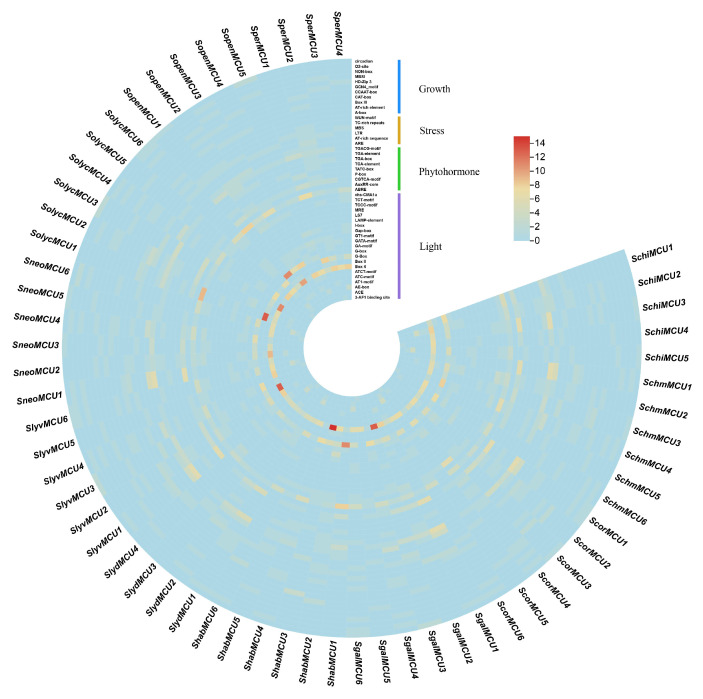
Analysis of cis-acting elements of MCU genes in Tomato genus.

**Figure 6 cimb-47-01021-f006:**
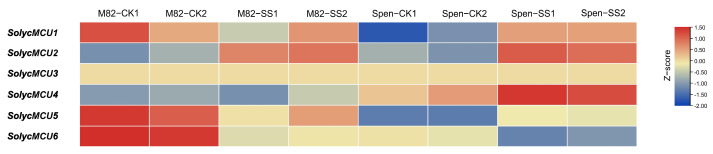
Transcriptome analysis of MCU genes under salt treatment. MCU gene expression levels in the roots of M82 and *S. pennellii*. CK represents the desert treatment, SS represents the salt treatment, and two biological replicates were used. Expression levels were log2-processed.

**Figure 7 cimb-47-01021-f007:**
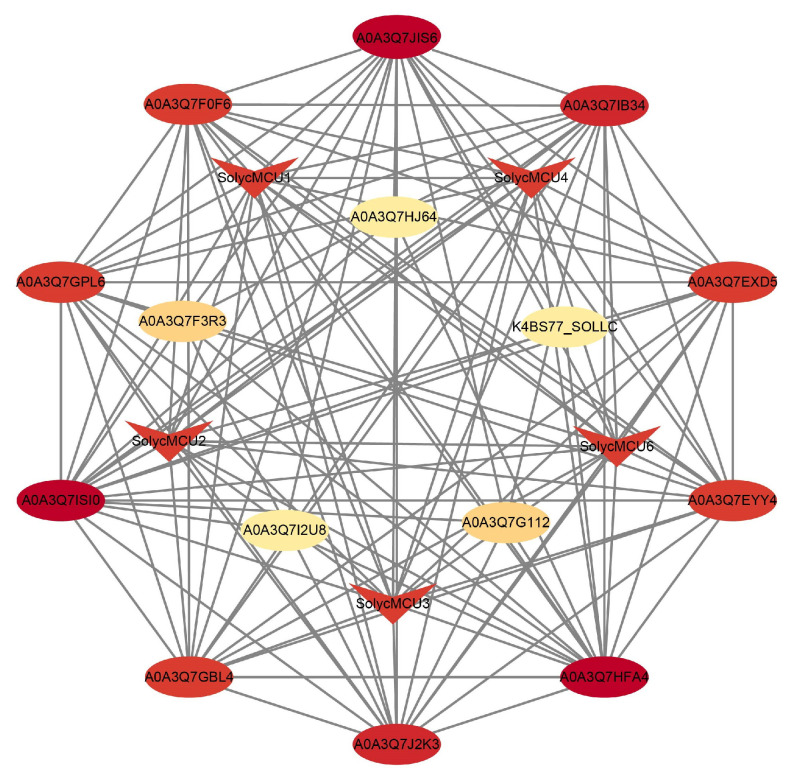
Prediction of Cultivated Tomato MCU Proteins Interactions. The degree algorithm is used to calculate the number of proteins with which a protein interacts. Darker colors indicate more interactions with other proteins. Rectangles represent MCU proteins.

**Figure 8 cimb-47-01021-f008:**
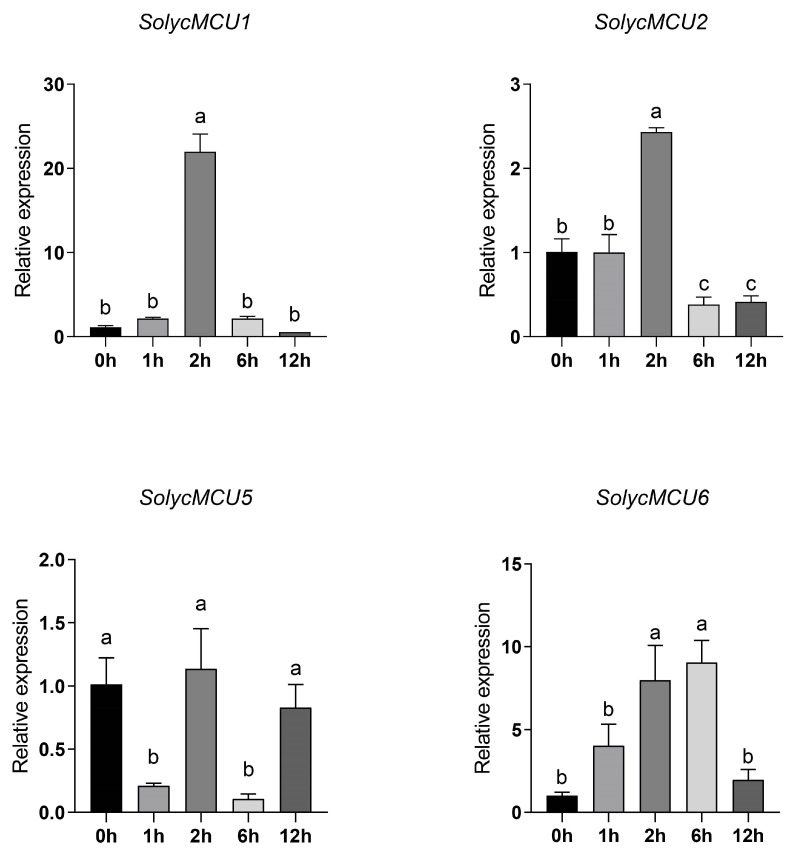
qRT-PCR Expression Analysis of the Cultivated Tomato *MCU* Genes. Different letters represent significant differences, shared letters represent no significant differences.

**Table 1 cimb-47-01021-t001:** Primer sequences for qRT-PCR experiments.

Gene	Forward Primer Sequence (5′ → 3′)	Reverse Primer Sequence (5′ → 3′)
*SolycMCU1*	AGAGAACTATGGGCTGGGCT	AGAAGGCTCTTTCGCAGTCC
*SolycMCU2*	TGTGATAAGGCGCGTGAAGT	TTGCACCAAGGATTCTGCCT
*SolycMCU5*	ATTCGAACGCCTAGCTCCAC	TGAGTTCATCGGTAACTCTCGAA
*SolycMCU6*	CGGACAGTTACATTGGTTGCG	TTCTTGTTTTCCCCACCCCC
*SlActin*	CAGGGTGTTCTTCAGGAGCAA	GGTGTTATGGTCGGAATGGG

**Table 2 cimb-47-01021-t002:** Physicochemical Properties of the Protein Encoded by the Heinz Tomato MCU Genes.

	Protein	Chromosome Location	Amino Acids	Theoretical pI	Molecular Weight	Instability Index	Aliphatic Index	GRAVY	Protein Secondary Structure			Subcellular Localization
									Alpha Helix	Beta Turn	Random Coil	
*Solanum* *lycopersicum*	SolycMCU1	chr2	349	9.16	39,715.17	56.35	83.5	−0.264	184	10	132	chlo_mito
	SolycMCU2	chr2	176	8.95	20,541.07	41.12	65.45	−0.273	104	4	46	mito
	SolycMCU3	chr3	303	9.34	35,949.08	39.92	75.25	−0.364	168	10	105	mito
	SolycMCU4	chr3	321	9.42	37,588.74	44.93	83.52	−0.336	172	11	117	mito
	SolycMCU5	chr4	351	9.46	40,810.63	53.54	88.03	−0.225	183	8	133	chlo
	SolycMCU6	chr6	365	9.43	41,922.88	33.81	96.08	−0.107	183	20	102	chlo

## Data Availability

The original contributions presented in this study are included in the article. Further inquiries can be directed to the corresponding authors.

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
