# Peer review of "Genome-Wide Identification of Mitochondrial Calcium Uniporter Family Genes in the Tomato Genus and Expression Profilings Under Salt Stress"

_cimb, 2025, doi:10.3390/cimb47121021_

Round 1
Reviewer 1 Report
Comments and Suggestions for Authors
Studying stress-related mechanisms in plants is a highly relevant topic, especially in today's rapidly evolving agricultural environment and unstable climate. This study attempts to unravel the mystery of the evolution and functional properties of tomato MCU genes.
The abstract clearly outlines the relevance, purpose, and results of the study and is written in clear, scientific language.
In the Introduction, the authors provide an overview of the topic. They also provide timely references to the literature on the topic. It has been established that MCU genes in plants are insufficiently studied. In the final section of the Introduction, the authors outline the purpose of their study and report their results. We propose excluding the results of this study from the Introduction, as they are presented in the Conclusion.
The Materials and Methods section is clearly written and well structured.
The Results section is also well presented and contains clear graphs for each analysis.
In the Discussion section, we would like to clarify a few points. The authors compare their results with studies on A. thaliana, Zea mays, and Oryza sativa, but do not provide references to these studies (442-446). Further in this section, the authors also do not provide references to studies, such as those on pears. Overall, we would have liked to see a comparison with other authors' work on this topic in this section.
The conclusion section is well-formulated. This is precisely why we see no point in presenting the results in the introduction.
Author Response
Comments 1: In the Introduction, the authors provide an overview of the topic. They also provide timely references to the literature on the topic. It has been established that MCU genes in plants are insufficiently studied. In the final section of the Introduction, the authors outline the purpose of their study and report their results. We propose excluding the results of this study from the Introduction, as they are presented in the Conclusion.
Response 1:Thank you for pointing this out. We agree with the reviewer’s constructive guidance.
Therefore, we have removed the specific findings in the Introduction section.
These changes can be found in the revised manuscript at Page 3, Paragraph 3, Lines 114-123.
The updated text in the manuscript is marked with a yellow background.
Comments 2: In the Discussion section, we would like to clarify a few points. The authors compare their results with studies on A. thaliana, Zea mays, and Oryza sativa, but do not provide references to these studies (442-446). Further in this section, the authors also do not provide references to studies, such as those on pears. Overall, we would have liked to see a comparison with other authors' work on this topic in this section.
Response 2: Thank you for bringing this relevant study to our attention. We agree with this comment.
Firstly, we have added the references at Line 465. Secondly, we have evaluated the suggested reference and determined that adding a comparison with the pear (Pyrus bretschneideri) MCU family significantly benefits the manuscript. Critically, this inclusion strengthens our discussion by extending the evolutionary context beyond model organisms. The finding that pear MCU genes also cluster into two subgroups and expanded via segmental duplication corroborates our results in tomato. Therefore, we have revised the Discussion section to cite this study and highlight these shared evolutionary patterns.
These changes can be found in the revised manuscript at Page 16, Paragraph 1, Lines 463-465.
The updated text in the manuscript is marked with a yellow background.
Reviewer 2 Report
Comments and Suggestions for Authors
This manuscript presents a comprehensive and timely genomic analysis of the MCU gene family across the tomato genus. The use of a pan-genome is a particular strength, providing evolutionary depth. The study is well-structured, moving from phylogeny to expression analysis, and the findings on the differential temporal expression under salt stress are intriguing. The work is potentially suitable for publication after minor revisions to address the points below.
- In the introduction, briefly define the knowledge gap, emphasizing the novelty of the pan-genome evolutionary approach. Provide a stronger, more explicit rationale for focusing on salt stress. Formulate and state a clear central hypothesis. Improve the logical flow by condensing the initial general sections and reducing repetition in the conclusion. Clarify the current understanding of the MCU complex in plants.
- The method of salt stress application is unclear and potentially problematic. The text states a "200 mM NaCl solution (pH 5.8, containing 1.5% agar)." It is not specified whether this is a hydroponic solution, a solid medium for rooting, or how it was applied to the 25-day-old seedlings (root drench, addition to agar plates?). Furthermore, the inclusion of agar in a treatment solution for soil-grown plants is unusual and needs justification.
- A critical control is missing. There is no mention of a parallel control group treated with a solution without NaCl (e.g., water or a nutrient solution). Without this, it is impossible to distinguish gene expression changes due to salt from those due to other factors like general shock or time of day.
- Cis acting element analysis method is well-described. The mention of a "clustering analysis" is good. Specify if this was hierarchical clustering and what method was used for the heatmap
- Lack of details in Transcriptome Analysis and Expression Profiling section. The public data is referenced as "[34]", but key details are missing. What was the duration of salt stress treatment in this dataset? What tissue? The term "integrated and analysed" is vague. Did you download raw reads and re-analyze them with a standard pipeline or use pre-calculated FPKM values? If the latter, how was batch effect between the two datasets accounted for? Why was this specific external dataset used instead of, or in addition to, the qRT-PCR data from your own experiment (2.1)? The connection should be explained.
- There is a significant error in table 1, the forward and reverse primer sequences for SolycMCU2are identical (TTGCACCAAGGATTCTGCCT). This is impossible for a functional primer pair and must be corrected immediately. All primer sequences must be re-checked for accuracy.
- The text states primers were designed in "conserved regions. qPCR primers must be designed in unique, gene-specificregions to avoid amplifying multiple MCU family members. The authors must provide evidence of specificity, such as a gel image, melt curve analysis, or a BLAST against the tomato genome confirming the primers are unique to each SolycMCU
- Line 225: The phrase "significant hydrophilic properties" is awkward. Suggest rephrasing to "All MCU proteins were predicted to be hydrophilic, as indicated by negative GRAVY values."
- Lines 238-240: The claim that gene absence in peruvianum and S. lycopersicoides is "attributed to gene loss events during tomato domestication" is incorrect and a major overinterpretation. Domestication refers to selection within the cultivated species (S. lycopersicum). The absence in these wild relatives is a lineage-specific gene loss event that occurred during the evolution of the genus, long before domestication. This must be corrected throughout the manuscript.
- Lines 255-264: The discussion of "artificial selection," "domestication," and "selective elimination" is not supported by the phylogenetic data and repeats the error from Comment 9. The tree shows evolutionary history, not domestication events. This speculative narrative should be removed. Please recheck.
- Lines 311-316: The claim that SolycMCU6"predates the monocot-dicot divergence" is a strong statement. The evidence (one collinear pair and phylogenetic position) is suggestive but not conclusive. To robustly support this, a phylogenetic analysis with basal angiosperms would be needed. The language should be toned down to "suggests an ancient origin" or similar.
- Lines 444-448 & 494-498: The discussion continues to propagate the unsupported claims about SolycMCU6predating monocot-dicot divergence and the link between gene loss and domestication. These sections need to be rewritten based on the corrected interpretations above.
- The discussion jumps between the root transcriptome data (M82 vs. pennellii) and the leaf qRT-PCR data (Heinz 1706) without explicitly acknowledging that these are from different tissues and genotypes. This limits direct comparability and should be noted as a limitation.
- Lines 526-527: The conclusion that evolutionary loss of SolycMCU3/4may reflect events during domestication is incorrect and must be removed, as per Comments 9 and 10. recheck
- The conclusions are generally sound but would be strengthened by incorporating the specific future directions mentioned in the discussion (e.g., tissue-specific expression analysis, functional validation via gene editing).
Round 2
Reviewer 2 Report
Comments and Suggestions for Authors
I have reviewed the revised manuscript. The authors have addressed all of my comments. I am satisfied with the revised manuscript and accept it in its current form.